# Breaking the Black-Box of Regional Resilience: A Taxonomy Using a Dynamic Cumulative Shift-Share Occupational Approach

**Francesca Silvia Rota [1,*], Marco Bagliani [2,3] and Paolo Feletig [4]**

[1] Research Institute on Sustainable Economic Growth, National Research Council, 10024 Moncalieri, Italy

[2] Department of Economics and Statistics "Cognetti de Martiis", University of Turin, 10125 Turin, Italy; marco.bagliani@unito.it

[3] Interdisciplinary Research Institute on Sustainability, University of Turin, 10125 Turin, Italy

[4] Socio-Economic Research Institute of Piedmont IRES Piemonte, 10125 Turin, Italy; feletig@ires.piemonte.it

[*] Correspondence: francesca.rota@ircres.cnr.it; Tel.: +39-339-223-5994

**Abstract:** In the European literature on the regional and local development, the concept of resilience has progressively gained momentum, eventually overcoming that of competitiveness and posing a critical challenge for the future of territorial studies and the territorialisation of the policy discourse. In the current economic turmoil, the success of an urban and regional economy relies more and more on its capacity to react to sudden shocks in a positive and evolutionary perspective, i.e., in its resilience. Nevertheless, as a recent analysis of the employment dynamics of Italian metro-regions in the period before and after 2008 has demonstrated that the existing taxonomies may be distant from reality and hardly communicable. The paper proposes a taxonomy of regional resilience based on the consideration of the region's capacity of both improving its employment rate during the pre-crisis period and overcoming the concurrent performance of the nation. Via a shift-share analysis of the employment in Italian metro-regions, the paper investigates the contribution of the sectoral structure of the local labour market in terms of economic resilience. The result is twofold: a geography of the dynamism of the territorial systems in Italy that diverges from some "classic" interpretative frameworks; a novel taxonomic approach to regional resilience.

**Keywords:** regional resilience; shift-share analysis; employment dynamics; sector composition; metro-regions

## 1. Introduction

The paper investigates how the conceptualisation of economic resilience can affect the design of territorial research methods and the building of regional taxonomies. From a geographical and regional perspective, this aim is meaningful because, for a long time, taxonomies have proven to be useful tools to detect territorial development trends and factors [1,2]. In regional studies, the identification of recursive evidence among territories can help the achievement of several goals, such as [3]: the up-scaling of assumptions and findings, the stratifying of samples of population and resources, the discovery of a selection of representative sites, and the framing of policies and reporting. Thus, taxonomic practices are an essential part of the work of academics [4].

Regarding the taxonomies of European regions, a preliminary overview of the literature is enough to realise how numerous they are [3]. They are recurrent above all in the studies that investigate the territorial patterns of competitiveness. For instance, in the realm of the innovation literature, regional scientists [5–7] and international territorial organisations such as the Organisation for Economic Co-operation and Development (OECD) [8] and the European Commission [9] contributed significantly

to the construction of regional taxonomies. While, in the context of a more holistic approach to regional competitiveness [10–12], emblematic taxonomies have been produced within the European Observation Network for Territorial Development and Cohesion (ESPON) initiative [13,14] with reference to different types of territorial attributes [3].

The aim of all these taxonomies is to divide the European territory into "convenient" groups of regions, connoted by homogeneity with respect to a given property (e.g., richness, competitiveness, innovation etc.). Some of them have also inspired important funding decisions by the public policies, becoming familiar to the broad society. The European Union (EU) Structural Funds, for instance, since the 1988 reform, used to classify the European regions of levels NUTS II and NUTS III according to their eligibility to a selection of priority objectives. More specifically, in the funding periods 1989–1993 and 1994–1999, four of the seven Structural Funds' objectives were reserved to selected regions: Objective 1, targeted to economic adaptation in regions lagging behind in economic development, was eligible only by NUTS II level regions with GDP per capita in PPP (purchasing power parity) below 75% of the Community average; Objective 2, dedicated to assist regions suffering from industrial restructuring, was reserved to NUTS III level (and smaller) regions with: unemployment rate above the Community average, a percentage share of Industry employment higher than the Community average, and a decline in the employment level of the Industry sector; Objective 5b, aimed at assisting rural regions with development problems, was addressed to the units smaller than NUTS III level with a low GDP per capita and two of the following statuses: high share of agricultural employment, low level of agricultural income, low population density and/or significant depopulation trend; Objective 6 concerned regions with very low population density of eight inhabitants per km$^2$ or less.

From a policy point of view, successful taxonomies are robust in method and broad in coverage, but also simple in conception and cross-analysis with other variables [3]. Ideally, the taxonomic exercise produces typologies that are neither too similar, nor conflicting. It mixes the rigorousness of the classification process with the clarity and transparency of the objectives and the capacity to substantiate the results of the analysis with the "experiential world" [1].

Often, however, the methods implemented to construct the taxonomies tend to be too sophisticated and biased by the context of the analysis and the availability of data [4]. It also happens in the studies on regional resilience, with some peculiarities.

In the resilience discourse, the need for a flexible adjustment to an increased number of emerging global challenges replaces that "survival of the strongest" approach that has characterised a large portion of the regional and urban discourse [15] since the first decades of the 20th century.

After the 2008 global economic crisis (and, more recently, as a consequence of the COVID-19 emergency), the world economy has profoundly changed. In a period characterised by frequent and dramatic turmoil, the success of the regional economy does not rely any longer on the search for techno-economic innovation, but also on the development of a mixed capacity of resistance, adaptation and creative exploitation of changes. The most important task has become the generation of a "fit-for-purpose" reaction, based on what Toynbee defined a "challenge and response" strategy [16]. Yet, this is also a definition of resilience. So, the concept of resilience increasingly accompanies that of competitiveness in the analysis of territorial development [17].

Secondly, as a consequence of the diffusion of an evolutionary approach to the conceptualisation of economic resilience [18–21], procedures of dynamic decomposition of the regional economic performance started gaining more and more attention [22,23]. In this conceptualisation, shocks represent acute modifications in the factors that regulate the functioning of the regional economic system, which are deeply contextualised in time and space. The idea is that shocks intertwine with the unfolding of broader processes of change and cause long-run adjustments [24] readable via the concepts of resilience and its corollaries (sensitivity, recoverability, resistance, antifragility). Table 1 identifies five renowned types of regions' reaction to shocks suggested by the literature, which have proved useful to classify the regional economies.

**Table 1.** Some main concepts of regional resilience and other forms of reaction to shocks.

| References | Typologies | Approach |
|---|---|---|
| Martin (2012) [22] | Resistant, Recovered, Re-orientated, Renewed | Evolutionary |
| Sensier et al. (2016) [25] | Resistant, Recovered, Not recovered in upturn, Not recovered not in upturn | Evolutionary |
| Martin et al. (2016) [23] | Most resilient (resistant and recovering), Least resilient (not resistant nor recovering) | Evolutionary |
| Blečić, Cecchini (2020) [26] | Resistant, Resilient, Antifragile | Planning |
| Equihua et al. (2020) [27] | Integer, Resilient, Antifragile | Ecosystemic |

Source: authors' elaboration.

The concepts listed in the first row of Table 1 derive from Martin's article "Regional economic resilience, hysteresis and recessionary shocks" [22]. Martin introduced the distinction between "resistance" (i.e., the capacity to contrast the adverse effects of the shock), "recovery" (i.e., the bouncing back from the immediate effects of the shock), "re-orientation" (i.e., region's structural realignment or adaptation) and "renewal" (i.e., when the growth path resumes to a pre-shock level). The second row refers to the results of a detailed analysis by Sensier et al., developed for the ESPON project "Economic Crisis: Resilience of Regions". According to these authors, some years after the occurrence of a crisis, the reaction of the regional economy can be of four types [25]: "resistant" if the region keeps on growing despite the shock; "recovered" if it overcomes rapidly from the effects of an initial contraction; "not recovered, in upturn", if the contraction produced by the crisis has already got to the trough and the region has started to grow again; "not recovered, downturn", if the trough still has to be reached. Similarly, the third taxonomy reported in Table 1, developed by Martin et al. [23], defines different typologies of regions based on the stage of the economic cycle experienced after the crisis (see Figure 1).

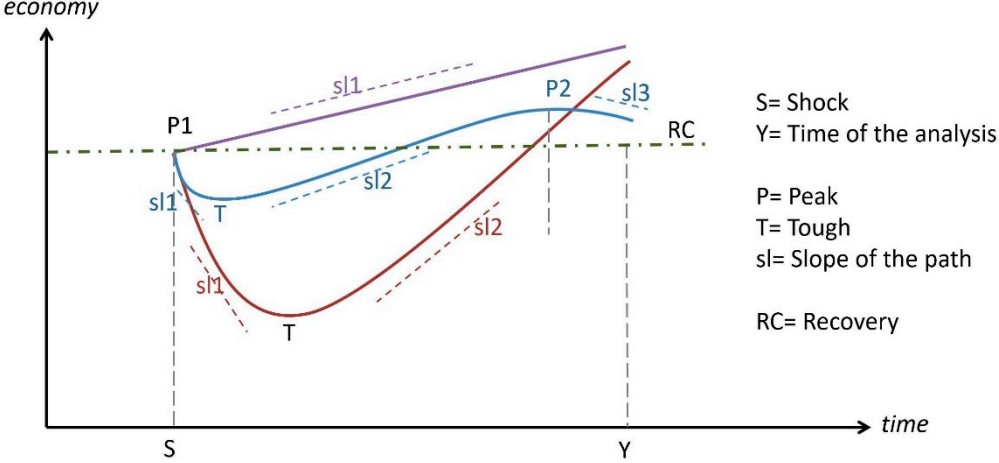

**Figure 1.** Stylised economic cycles after a shock. Source: authors' elaboration from Sensier et al. [25] p. 137.

The two taxonomies reported at the bottom of Table 1 report the region's reactiveness within a hierarchical discourse. Blečić and Cecchini [26], in particular, define "resilience" as a complement of a more comprehensive condition of regional recovery called "antifragility". The ability to withstand the adverse events determined by the shock, without being damaged disproportionately, is instead considered a complement of the resilience, described as "robustness". The study by Equihua et al. [28] too deals with the category of antifragility, but in reason of its being a substitute (not an attribute) of robustness and resilience. In the authors' words: "while resilient/robust systems are merely perturbation-resistant, antifragile structures not only withstand stress but also benefit from it".

As we can see, the taxonomies underlying an evolutionary approach pay specific attention to the capacity of regions to start growing again and recover past economic performance. The concepts concerning a planning and ecosystemic approach, instead, describe the status of the region at the time of the analysis. All the cases listed in Table 1, however, show a common trait in evaluating regional resilience in absolute terms, i.e., without considering what occurs out of the borders of the region. In this paper, we adhere to an evolutionary perspective, recognising, however, more attention to the way the regions perform with respect to the other regions. The amplitude and the duration of the recovery are not significant per se, but to the extent that they overcome in size and pace the average/aggregate recovery. How the crisis impacts on the national economy, in particular, is recognised to have a dramatic influence on the resilience, resistance or antifragility of regions.

Consistent with this, we differentiate the concept of resilience according to an absolute and relative dimension. Absolute resilience refers to the capacity of the region to safeguard its initial economic performance, i.e., the performance it had before the crisis. Relative resilience compares the type and intensity of the reaction of the region to the reaction of the nation. In a perspective of absolute resilience, the most crucial information is the direction and slope of the growth path (see Figure 1). The scholars that follow this approach thus tend to have little consideration for resistance and sensitivity as dimensions of resilience [25,29]. Conversely, in a perspective of relative resilience, which also considers the sign of the national trend before and after the crisis, regional resilience results in the capacity of either avoiding sub-optimal growth rates or starting a path of growth better than the national one [28].

Following Martin et al. [23], our impression is that the reactions to the 2008 crisis are so complex and diverse that a shift away is required from territorial typologies uniquely based on the criteria under and over the national threshold. These typologies can appreciate the contribution of the structural and local conditions just loosely [30–34]. For this task, there is the need for a different taxonomy, capable of interpreting the different economic cycles that follow the shock (see Figure 1) in the light of the possible combinations of local conditions, ultimately highlighting different territorial patterns of resilience. It is mostly on these patterns that regional policies can act to create and reinforce regional resilience processes [35]; yet, their identification remains a controversial and challenging task, especially with respect to the already existing territorial typologies and the perceived state of the art of the relations among regions.

The rationale of the paper refers to this open issue. It aims at building an easy-to-use methodology of regional analysis, based on the concept of resilience and coherent with the complexity [36] and the actual dynamism of territories [30].

Our study contributes to this end by proposing a novel taxonomy based on an original methodology in which the different territorial patterns of resilience emerge from the consideration of the regional economic performance compared with the national one, as well as the overall capacity of the region to maintain the employment levels, and the sectoral composition of the regional economy. For some authors, sectors have proven to play an essential role in influencing the different sensitivity levels demonstrated by countries and regions of the world concerning the global economic crisis of 2008 [29,37]; i.e., they contribute to the moulding of different territorial patterns of resilience [32,38–40].

The paper aims to construct a taxonomy of the regional patterns of economic resilience and to empirically identify them regarding the response of Italian metro-regions to the 2008 crisis.

Section 2 describes the data and methodology of the analysis. Section 3 illustrates the proposed taxonomies of resilience to the crisis. Section 4 completes the discussion with the case of Italian metro-regions and develops the concluding remarks.

## 2. Materials and Methods

### 2.1. A Three-Step Taxonomic Approach

In line with evolutionary theories, regional resilience is considered here as a continuous process of self-adjustment through feedback and learning mechanisms, which do not allow the achievement of a stable equilibrium condition [18,19,41–45].

The first methodological consequence of this approach is the necessity of a long-term perspective. According to Boschma [20], this is essential to reconstruct the region's ability to reconfigure its industrial, technological and institutional structure. However, there is no agreement on the physiological duration of the recovery of a regional economy. According to Hill, four years are enough for the regional self-organising response in order to emerge [21]. Here, we prefer to consider a larger timespan, consisting of the eight years before and after the starting of the crisis in 2008.

The second methodological consequence concerns the spatial scale of analysis. Resilience studies are conducted at almost all the territorial units. Here, we have assumed the metro-regional scale of analysis to be as in the literature; it has proven to be highly functional to discern development gaps among and within the European territories, measured using the employment variable [30,35,46]. The third and last methodological consequence regards the influence of local conditions on the region's ability to create autonomous responses to shocks [47]. Evolutionary theories underline the role played by place and context in creating a specific system of cultural, social and institutional contingencies, which limit the options of regional development within a range of possibilities not too far from the initial trajectory [20]. This condition of path-dependency explains why, in the face of the shock, some economies renew themselves, while others decline [48]. A path-dependent approach, claiming for considering the influence that the sector composition of the economy exerts on regional resilience, has thus become customary in the literature [18,23,25,38,49]. Following these examples, the paper identifies in the dynamic cumulative calculation of the shift-share effects on employment the best technique to distinguish (and quantify) the contribution given by the regional sectors, and by the regional competitive advantage, to the deviation of the regional growth path from the national one. See Lahr and Ferreira [50] for a discussion of the potentialities of shift-share. The methodology proposed here uses the annual employment growth rate as the basic variable to measure regional resilience and entails three complementary steps:

- The calculation of the regions' trends in employment before and after the crisis, and in comparison with the national one;
- The classification of the regions according to six territorial typologies of resilience that reflect: whether the employment rates in the pre-crisis and post-crisis periods were over or below the national ones; whether the number of employees increased or not after 2008;
- The quantification of the influence of the sectoral distribution of employment, via a dynamic formulation of the shift-share methodology.

### 2.2. The First Step: Using Regional Trends and Occupational Capacity to Propose a New Taxonomy

The first phase, inspired by the sensitivity index proposed by Martin [22,24,51], estimates the regional trends in employment, both before and after the crisis, and compares them to the national trend. In particular, we calculated the trends with the expression:

$$(E_{ir}^{t_{0+h}} - E_{ir}^{t_0})/E_{ir}^{t_0} \tag{1}$$

where $E_{ir}^{t_0}$ indicates the employment of sector $i$ and region $r$ and $t_0$ and $t_{0+h}$ are the initial and final year of the chosen interval of time.

Martin's sensitivity index is calculated as a ratio between percentage variation in employment in a region and the respective variation in the country as a whole, as expressed by the following expression:

$$\beta_r = (\Delta E_r / E_r) / (\Delta E_r / E_r) \tag{2}$$

where $\Delta E_r / E_r$ is the percentage variation in employment of region $r$ and $\beta$ is the "sensitivity index". Differently, in our model, the employment trends before and after the crisis are plotted on a Cartesian graph: period 2000–2008 on the *x*-axis; period 2008–2016 on the *y*-axis. Since the focus is on relative economic performance of regions, i.e., their performing either better or worse than the nation in terms of employees growth, the origin of the graph, where the axes x and y intersect, is not placed on (0, 0), but corresponds to the national values.

To capture the information of the yearly variation of the regional employment, which does not emerge from the calculation of the average trend, two more metrics are introduced, called *absolute occupational capacity* and *relative occupational capacity*. As explained further in Appendix B, from a mathematical point of view, the absolute occupational capacity for the period between year $t_0$ and $t_{0+h}$ represents the discrete version of the integral of the cumulative sum defined in Equation (5). The *absolute* occupational capacity thus sums, for each year of the time interval, the difference of employees with respect to year $t_0$. To facilitate the comparison between regions, we expressed this new metric in percentage terms, dividing it by $E_{ir}{}^{t_0}$ (the number of the employees at year $t_0$).

The relative occupational capacity of a region is calculated as the difference between its absolute occupational capacity and the occupational capacity of the nation.

The regional relative occupational capacities of metro-regions, for the entire period 2000–2016, are reported on the Cartesian graph (see Section 3.1) as circles of different magnitudes and colours.

### 2.3. The Second Step: A New Taxonomy of Regional Response to the Crisis

As the second phase of our study, we have developed a methodology based on the Cartesian diagram built in the first step. The schemes in Figure 2 exemplify the criteria adopted in the setting up of the taxonomic methodology. From a graphic point of view, the deviation from the *y*-axis registers the distance of the regional growth rate from the national one in the post-crisis period.

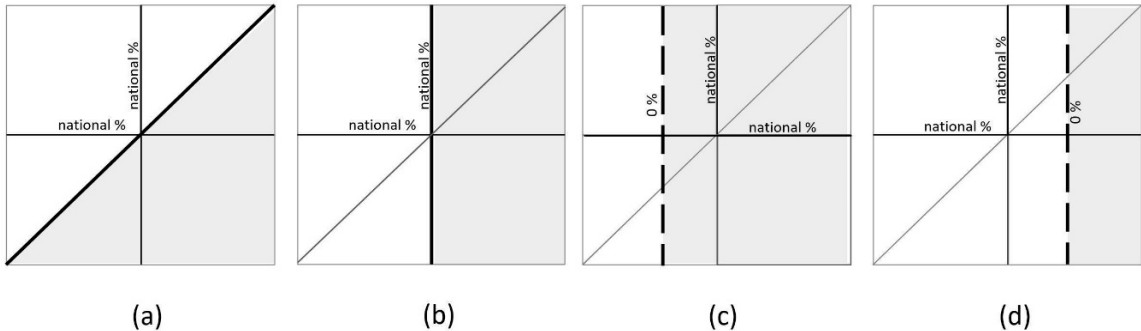

**Figure 2.** The setting up of the taxonomic methodology. In grey are the sectors meeting the conditions: (**a**) regional post-crisis rates above the regional pre-crisis ones; (**b**) regional post-crisis rates above the national post-crisis ones; (**c**) positive post-crisis regional rates, in a context of positive national rate; (**d**) positive post-crisis regional rates in a context of a negative national rate. Axes *x* and *y* report the regional trends in employment in the post- and pre-crisis periods, respectively, while the dashed line represents the value of zero growth in the post-crisis period. The horizontal dashed line, indicating the value of zero growth in the pre-crisis period, is not represented in the graphs, as it is not utilised for the construction of the taxonomy.

In order to distinguish different territorial typologies of resilience, we consider the following conditions:

(1)     The region has improved its relative performance (calculated as trends in employment) in the post-crisis period. This condition corresponds to the portion of the graph on the right of the main diagonal (see Figure 2a);

(2)     The region has performed better than the nation in the post-crisis period. This condition corresponds to the portion of the graph on the right of the *y*-axis (see Figure 2b);

(3)     The region has registered in the post-crisis period a positive variation of its employment levels (growth). This condition corresponds to the portion of the graph on the right of the dashed line representing the value of zero growth in the post-crisis period. This line can occupy a different position in the Cartesian scheme according to the fact that, in the same period, the nation has gained or lost employment (see the dashed line in Figure 2c,d, respectively).

The joint consideration of the first two criteria allows for the identification of six types of response to the crisis, graphically corresponding to the six areas identified in Figure 3:

- The first type (1) corresponds to the regions characterised by solid economies, fuelled by employment rates that are higher than the national one, both in the pre-crisis and in the post-crisis period, and that improve their performance in the period 2008–2016.
- The second type (2) identifies the regions whose employment rates are higher than the nation in both of the considered periods, although resized in the passage to the post-crisis period.
- The third type (3) contains regions characterised by growth rates that are worse than the national one in the pre-crisis period and better in the post-crisis one. These regions have also improved their performance in the post-crisis period, eventually emerging for their capacity to react to the crisis proactively.
- The fourth type (4) corresponds to the top left sector of the graph. The regions of this group are fragile and vulnerable as they failed in maintaining the comparative advantage, they held in the pre-crisis period in terms of employment dynamism. In the period 2008–2016, their occupational capacity decreased to a level lower than the national one.
- The fifth type (5) corresponds to the portion of the bottom left sector, right of the diagonal. The regions in this area are characterised by weak economies, with employment trends lower than the national average both in the pre-crisis and in the post-crisis period. Nevertheless, their employment trends improved after 2008, signalling a certain capacity of reaction.
- The sixth type (6) embraces the regions that fall in the bottom left sector, left of the diagonal. This area experienced the worst situation of all, i.e., employment rates that are always below the national average and resize in the passage to the post-crisis period.

However, the identification of these six types of response to the crisis is not the final stage of our taxonomy. In order to take into account the complete information available, also the third criterion (shown in Figure 2c,d) has to be considered. The fulfilment of all the three criteria is obtained by the superposition of the six types of response to the crisis shown in Figure 3 with the schemes of Figure 2c (if the nation has gained employment in the post-crisis period) and Figure 2d (if the nation has lost employment in the post-crisis period). The results of such a superposition lead to the identification of the taxonomies of regional resilience shown in Figure 4. The former of the two graphs in the figure refers to the hypothesis that the variation of the national employment in 2008–2016 is positive; the latter to the case it is negative variation. To the best of our knowledge, these schemes provide a novel, practical, and easy to reproduce tool for the identification of the resilient regions.

We outline that the use of this third additional condition allows one to depict the territorial reactions to the crisis better. More specifically, the regions that stand on the right of both the *y*-axis and the dashed line emerge as endowed with an additional attribute, that, following Martin et al. [23], can be defined as *resistance*: the property of a region of maintaining positive growth rates of the economic variables, despite the crisis.

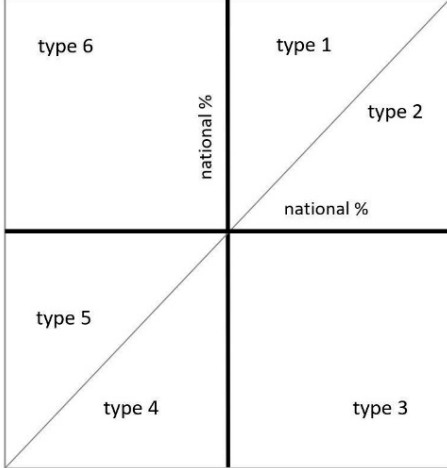

**Figure 3.** The six types of the regional response to the crisis that derive from the joint use of conditions 1 and 2 (see the beginning of Section 2.3). Axes $x$ and $y$ report the regional trends in employment in the post- and pre-crisis periods, respectively.

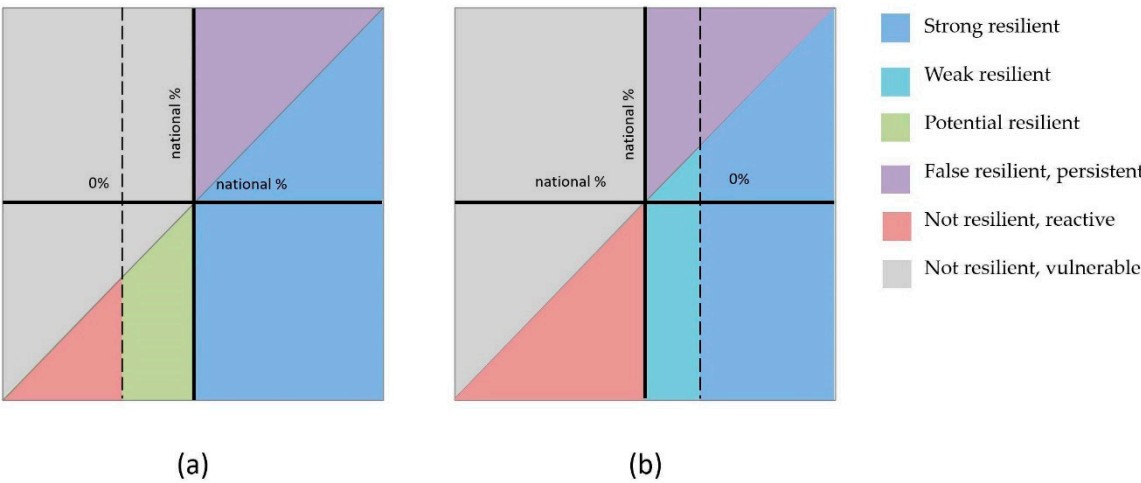

(a)             (b)

**Figure 4.** The taxonomy identifies the typologies of regional resilience in cases where the variation of the national employment levels in the post-crisis period is either positive (**a**) or negative (**b**). Axes $x$ and $y$ report the national rates in the post- and pre-crisis periods, respectively, while the dashed line represents the value of zero growth in the post-crisis period. As mentioned regarding Figure 2, the horizontal dashed line, indicating the value of zero growth in the pre-crisis period, is not represented as it is not utilised for the construction of the taxonomy.

As a final result, the proposed taxonomy identifies six different typologies of a regional response to the crisis (see Figure 4a,b):

- *Strong resilient*. These regions present growth rates higher than the national ones, both in the pre-crisis and in the post-crisis periods. In addition, they register an improvement in absolute terms of the occupational performance in the post-crisis period, which can be evidence of reaction. Finally, their economy shows clear features of resistance, as demonstrated by the fact that the growth rate in the post-crisis period is positive in sign.
- *Weak resilient*. These regions have growth rates higher than the national ones in the pre-crisis period as well as in the post-crisis one, and they register in the post-crisis improvement of their performance in absolute terms, which can be considered evidence of dynamism/reaction capacity. However, concerning the strong resilient, these regions present a negative growth rate of the occupational performance, so as they are defined as weak resilient.

- *Potential resilient*. These regions improve in the post-crisis period their performance in absolute terms, which can be a clue of dynamism, and they can also rely on a good resistance to the crisis that allows them to maintain positive growth rates. Nevertheless, in the post-crisis period, their performance is negative and below the national one, so they cannot be defined as fully resilient, but just as potentially resilient.
- *False resilient, persistent*. These regions are characterised by economic resistance. They register positive relative performance in both the pre-crisis and the post-crisis period (since located in the upper-right sector), and their growth rates are always higher than the national average (since located to the right of the vertical dashed line). Nevertheless, the occurrence of declining performances in time (since located to the left of the diagonal) suggests a condition of poor reaction.
- *Not resilient, reactive*. What characterises these regions is the fact that, despite being economically fragile (growth rates are negative and lower than the national ones in all the considered periods), they are located to the right of the diagonal, which means they had a positive reaction to the crisis.
- *Not resilient, vulnerable*. In this sector, regions do not meet any of the criteria of resilience: both in the pre-crisis and in the post-crisis periods, they present growth rates worse than the national ones, and worsen their occupational performance after 2008; therefore, they are considered not resilient in a vulnerable way.

*2.4. The Third Step: The Quantification of the Sectoral Influence on Resilience*

The third phase of the study aimed at investigating why the regions faced the 2008 economic crisis differently. In this phase, we analysed the contribution of the different economic sectors to the emergence of a specific condition of regional resilience or fragility. The method we adopted is a dynamic formulation of the well-known shift-share methodology [52–54] that allows one to describe the economic behaviour of a sample of local systems (in this study the metro-regions defined by Eurostat), comparing them to the national dynamic.

Considering the variation in time of a given economic variable (the employment), the dynamic and cumulative equation of the shift-share decomposes it into some partial effects. Applied separately for the years before and after the onset of a shock, it provides a substantial help for the detections of similar territorial patterns of reaction, based on the observed recurrences and similarities.

In our study, the dynamic-cumulative method described and already used in Bagliani et al., 2020 [30,55], for the analysis of the competitiveness of Italian metro-regions, is proposed in order to build an innovative taxonomy based on the concept of resilience. Our methodology originates from the Esteban-Marquillas formulation of the shift-share [53], that, considering a given time period between year $t_0$ and $t_{0+h}$, splits the variation of occupation $E$ into four different effects, according to the equation:

$$\Delta E_{ir}{}^{t_0} = E_{ir}{}^{t_{0+h}} - E_{ir}{}^{t_0} = NGE_{ir}{}^{t_0} + IME_{ir}{}^{t_0} + CSE_{ir}{}^{t_0} + AE_{ir}{}^{t_0} \tag{3}$$

where $i$ indicates the economic sector, and $r$ the region. The effects are defined by the following expressions:

$$NGE_{ir}{}^{t_0} = E^*_{ir}{}^{t_0} \, g_{iN}{}^{t_0}$$

$$IME_{ir}{}^{t_0} = (E_{ir}{}^{t_0} - E^*_{ir}{}^{t_0}) g_{iN}{}^{t_0} \tag{4}$$

$$CSE_{ir}{}^{t_0} = E^*_{ir}{}^{t_0} \, (g_{ir}{}^{t_0} - g_{iN}{}^{t_0})$$

$$AE_{ir}{}^{t_0} = (E_{ir}{}^{t_0} - E^*_{ir}{}^{t_0}) \, (g_{ir}{}^{t_0} - g_{iN}{}^{t_0})$$

where the subscript $N$ indicates the national value and:

$E^*_{ir}{}^{t_0} = (E_r{}^{t_0})(E_{iN}{}^{t_0}/E_N{}^{t_0})$ designates homothetic employment, i.e., the number of employees that region $r$ would register in sector $i$ if it had the same national sector composition;

$g_{iN}{}^{t_0} = (E_{iN}{}^{t_{0+h}} - E_{iN}{}^{t_0})/E_{iN}{}^{t_0}$ indicates the growth rate of sector $i$ at the national level, in the period between year $t_0$ and year $t_{0+h}$;

$g_{ir}{}^{t_0} = (E_{ir}{}^{t_{0+h}} - E_{ir}{}^{t_0})/E_{ir}{}^{t_0}$ describes the growth rate of sector $i$ for region $r$ in the period between year $t_0$ and year $t_{0+h}$;

$E_r{}^{t_0} = \Sigma_i\, E_{ir}{}^{t_0}$ and $E_N{}^{t_0} = \Sigma_i\, E_{iN}{}^{t_0}$, respectively, indicate the total number of employees in region $r$ and in the nation.

Consistent with the established use, we identify the four effects that follow [52–54]:

- The *National Growth Effect* (*NGE*) quantifies the growth that the region would have registered if it had the same national sectoral composition and grew at the average national rates. It represents the term that allows the comparison of the dynamics observed by the region with the national average;
- The *Industry Mix Effect* (*IME*) indicates the contribution of the sectoral composition of the region compared to the national one. It estimates whether the region is specialised in sectors that, on a national scale, are experiencing a phase of growth or crisis;
- The *Competitive Share Effect* (*CSE*) measures the different capacity of regional sectors to create employment compared to that of the same sector at the national level;
- The *Allocative Effect* (*AE*) indicates the competitive efficiency of national sectors, that is, whether regional specialisation is more distributed in sectors that are more or less efficient than the national average in creating new jobs.

To allow a more detailed examination of the temporal variation of the four effects, we used the dynamic-cumulative formulation of the shift-share, proposed and discussed in Bagliani et al. [30]. Considering the above-mentioned interval of $h$ years between $t_0$ and $t_{0+h}$, we calculate, for each year $k$ included in the interval, the quantity $CS_{ir}{}^{t_0-t_{0+k}}$ representing the *cumulative sum* from $t_0$ to $t_{0+k}$, of the annual shift-share effects obtained by Equation (3). The corresponding formula is:

$$CS_{ir}{}^{t_0-t_{0+k}} = \Sigma_m\, \Delta E_{ir}{}^m = \Sigma_m\, (E_{ir}{}^{m+1} - E_{ir}{}^m) = \Sigma_m\, NGE_{ir}{}^m + \Sigma_m\, IME_{ir}{}^m + \Sigma_m\, CSE_{ir}{}^m + \Sigma_m\, AE_{ir}{}^m \quad (5)$$

where $k$ can assume values between 1 and $h$ and indicates in which year, after $t_0$, the sum stops, while $m$ varies between $t_0$ and $t_{0+k}$. This cumulative sum accounts for the total amount of employees added or lost during the time interval between $t_0$ and $t_{0+k}$, with respect to year $t_0$ (see also Appendix B). Thanks to this methodology, it is possible to analyse the development over time of the shift-share effects and not just the sum over the whole interval, as the original shift-share formulation does [53].

Again, in order to make the comparisons between regions easier and more meaningful, we express the cumulative sum in percentage values dividing Equation (5) by $E_{ir}{}^{t0}$. Percentage cumulative sum is used to assess, in Sections 3.2–3.5 the sectoral influence on resilience.

The number of sectors in the shift-share analysis is not a priori determined. Martin et al. [23], for instance, run a fine-tuned 25-sector disaggregation. Artige and van Neuss [56] in their analysis of the Belgian manufacture used data from 14 sub-sectors. In this study, we propose to consider 11 economic sectors, resulting from the NACE codes listed in Appendix A.

## 3. Results

### 3.1. Territorial Patterns of the Resilience of the Italian Metro-Regions

This section describes the results of the application of the taxonomic procedure described in Section 2 to the Italian metro-regions. More specifically, we consider the 21 metro-regions identified in Italy by Eurostat (see Appendix C).

Following the methodology described in the first and second step of the procedure (see Sections 2.2 and 2.3) we obtain the graph of Figure 5, that allows for the identification of the regional typologies of resilience via the taxonomy proposed in Section 2.3. Figure 5 reports the relative economic performance of the Italian metro-regions in the 2000–2008 period (*y*-axis) and the 2008–2016 period (*x*-axis), and the regional relative occupational capacity. The variable of relative occupational capacity, in particular, is represented by the symbol of a sphere, whose size and colour are representative of the value and

the sign of the variable, respectively. Furthermore, the vertical dashed line indicates the value of zero growth in the post-crisis period.

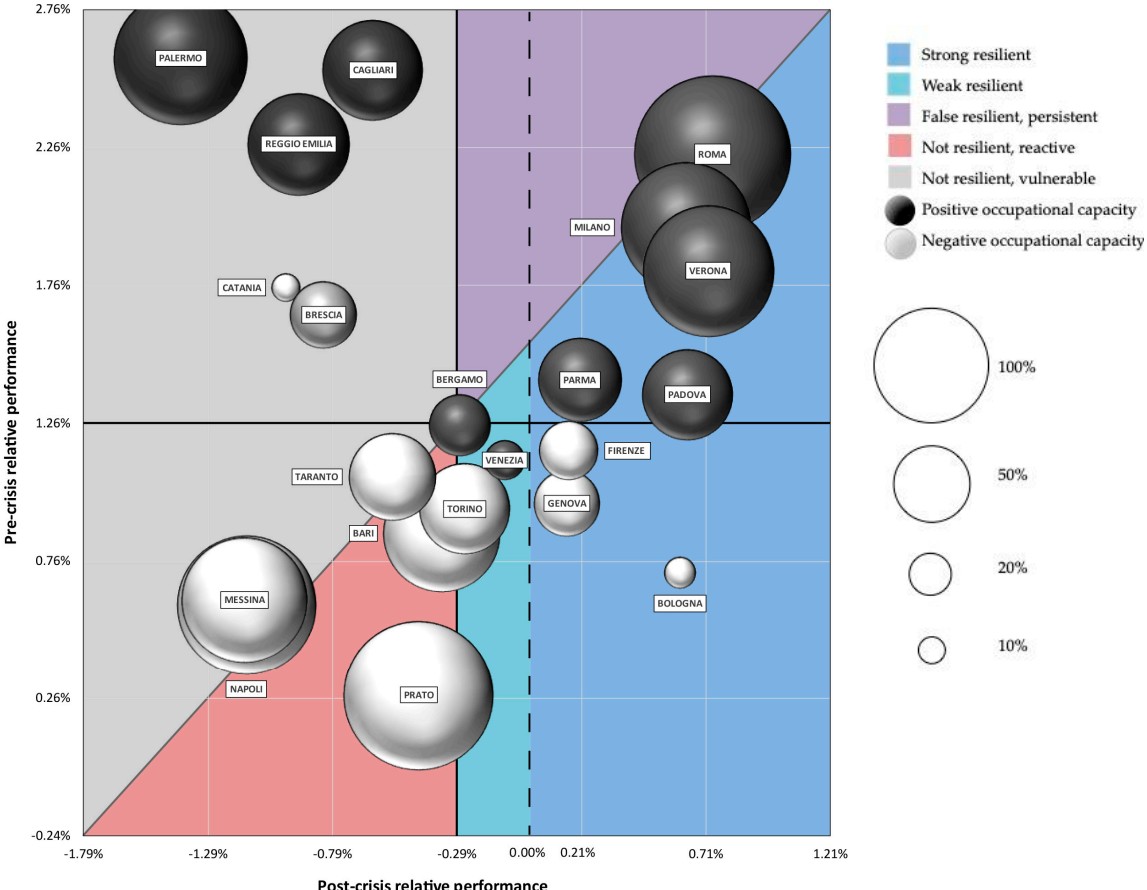

**Figure 5.** Italian metro-regions and their position concerning the trend in employment in the 2000–2008 period (*y*-axis) and the 2008–2016 period (*x*-axis). The intersection of the axes is not placed on (0, 0) but on the national values of the post- and pre-crisis periods. The dashed line represents the value of zero growth in the post-crisis period. The circles of different magnitudes and colours represent the relative occupational capacity of the regions (see Section 2.2).

The analysis of the Italian case shows an average national growth rate negative in the post-crisis. This means that our case study falls in the typology described by Figure 4b.

The results draw a novel geography of the economic performances of the Italian territories. Following the scheme of Figure 4b, we identify eight *strong resilient* metro-regions (Bologna, Firenze, Genova, Milano, Padova, Parma, Roma, and Verona) and three *weak resilient* metro-regions (Bergamo, Torino and Venezia). None of the analysed cases falls into the typology of the *persistent* regions, while two of them result *not resilient, but reactive* (Bari and Prato). The remaining eight metro-regions are classified as *vulnerable* (Brescia, Cagliari, Catania, Messina, Napoli, Palermo, Reggio Emilia, and Taranto). In addition, the spheres of Figure 5 also report the information of the relative occupational capacity, from which we can further distinguish the metro-regions, according to the way their resilience combines with the pre-crisis employment levels. For instance, Roma and Milano belong to the group of the strong resilient and are also characterized by a positive consistent occupational capacity. Conversely, the occupational capacity of Firenze, Genova and Bologna is negative, despite their resilience. Among the weak resilient metro-regions, Venezia and Bergamo have a positive occupational capacity, while Torino has a negative one. Additionally, in the group of the vulnerable metro-regions, the results for this variable are diversified:

Palermo, Reggio Emilia and Cagliari register a positive performance; Catania and Brescia slightly negative; Taranto, Bari, Napoli and Messina highly negative.

About the third step, for practical reasons, the shift-share analysis with the detail of the sectoral contribution to the occupational rates is not reported in a yearly time series, but as the partial sum of the whole pre-crisis (2000–2008) and post-crisis (2008–2016) periods.

In the following sections, the metro-regions of each typology are described according to the results of the shift-share analysis of the employees for each economic sector in the periods 2000–2008 and 2008–2016, as calculated by Equation (A1) in Appendix A. The aim is to point out recurrences and divergences in the way the economic sectors influence the resilience of the Italian metro-regions.

### 3.2. Strong Resilient Italian Metro-Regions

The dark blue sector of Figure 5 identifies the group of the most resilient territories. It includes the two largest metro-regions of Italy, which are also Metropolitan cities (Milano and Roma), three smaller Metropolitan cities (Bologna, Firenze, and Genova) and three medium-sized metro-regions (Padova, Parma, and Verona). Due to their employment stability, they can be considered the strongest and driving regions of the national economy. Roma, Milano and the northern metro-regions stand out with positive performances in all the variables considered in the analysis (pre and post-crisis relative employment performance, resistance and occupational capacity). Conversely, the smaller Metropolitan cities of Genova, Firenze and Bologna show negative values in the pre-crisis dynamic, and they also register a negative relative occupational capacity (quantified, respectively, in −22.5%, −17.8% and −5.2%). This latter condition, in particular, derives from the fact that, at the time the crisis started, these metro-regions had already suffered from a consistent loss of employment that was just partly recovered after 2008.

Before the crisis, the employment growth rates of these metro-regions were beneath the national average; and, in the post-crisis period, they registered a smaller relative occupational capacity. Focussing on the results of the shift-share analysis (see Figure 6), the dramatic fall emerges, occurring after 2008, of the propelling role of the national level. It is evident above all in Milano and Roma, where the negative occupational dynamic of the nation (NGE effect) in the post-crisis period caused a loss of, respectively, 352,000 and 347,000 employees. However, the regions of this group succeeded in rebalancing the losses thanks to the overall competitiveness of the regional system (positive CSE effects are present in Milano, Padova, Verona, and Parma) and the favourable sectoral organisation of the local economy (testified, for instance, by the positive IME effect of Roma).

As Figure 6 shows, sectors with common dynamics in the post-crisis period are:

- Construction (f): in this sector the occurrence, after 2008, of widely negative national effects (NGE) has determined a shift from expansion to contraction, and losses that vary from −5000 to −10,000 employees. The only exception is Genova, where a substantial competitive effect (CSE) has determined an increase of approximately 1300 employees;
- Retail and Logistic (g–i): in all the regions, but Genova and Padova, this sector registers at the end of the post-crisis period an increase in employment, fuelled by the competitive effect (CSE), i.e., by the regions' capacity (strong and evident above all in Bologna) of supporting the occupation in the Retail and Logistic activities in a situation in which the sector suffered significantly from the crisis;
- Technical and Scientific Services (m–n): with the only exception of Verona, these are the sectors that grew most in the pre-crisis period. In addition, the persistence of a positive, although reduced, national effect (NGE) allowed them to maintain a trend of employment growth;
- Public Administration Services (o–q): independently from what happened before 2008, in the 2008–2016 period, in this sector, all the regions showed positive competitive effects that helped them to maintain (or increase) the employment levels. Remarkably, the most positive increases occurred in Milano, Roma (approximately +7800 employees both) and Padova (+7500 employees).

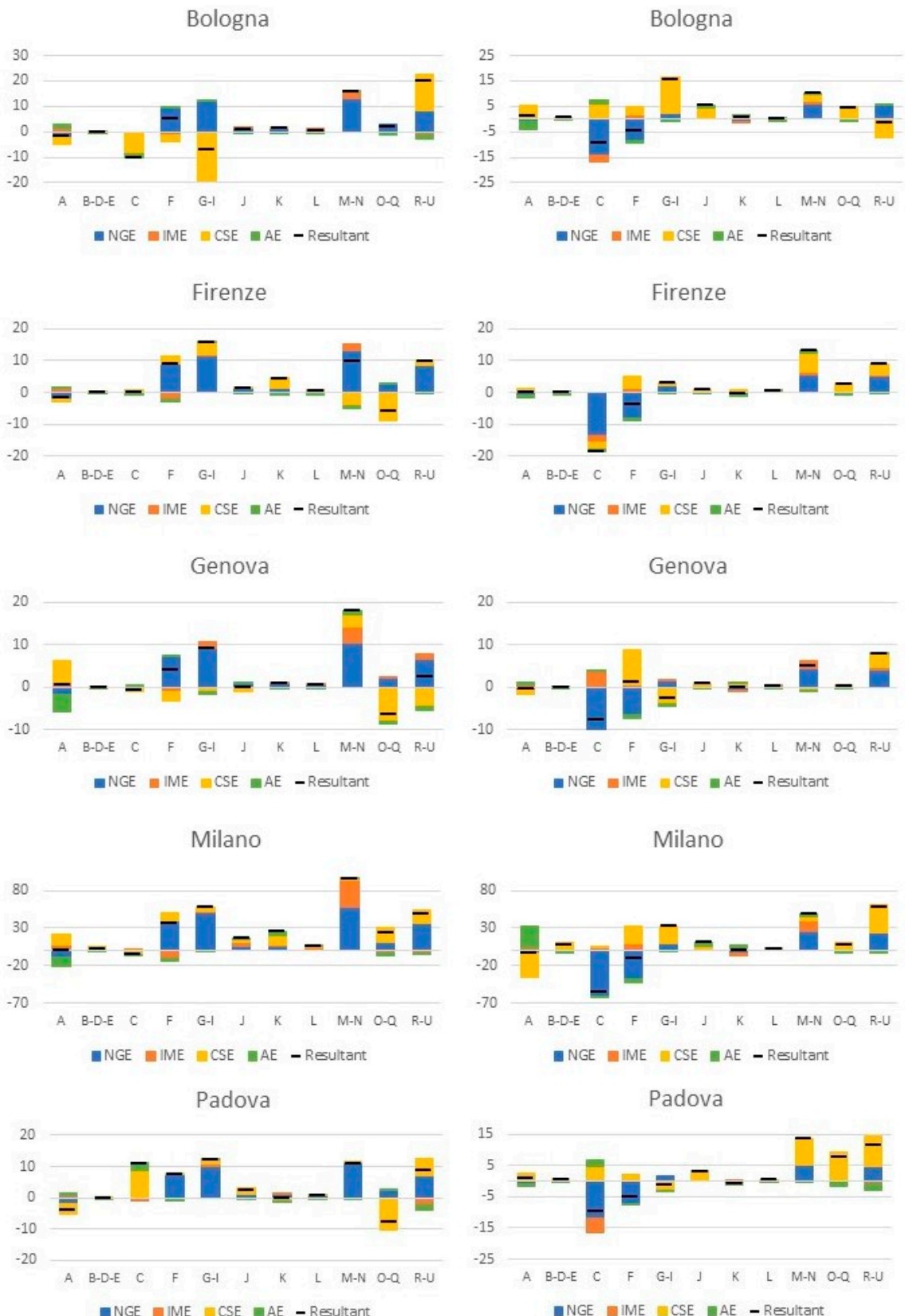

**Figure 6.** *Cont.*

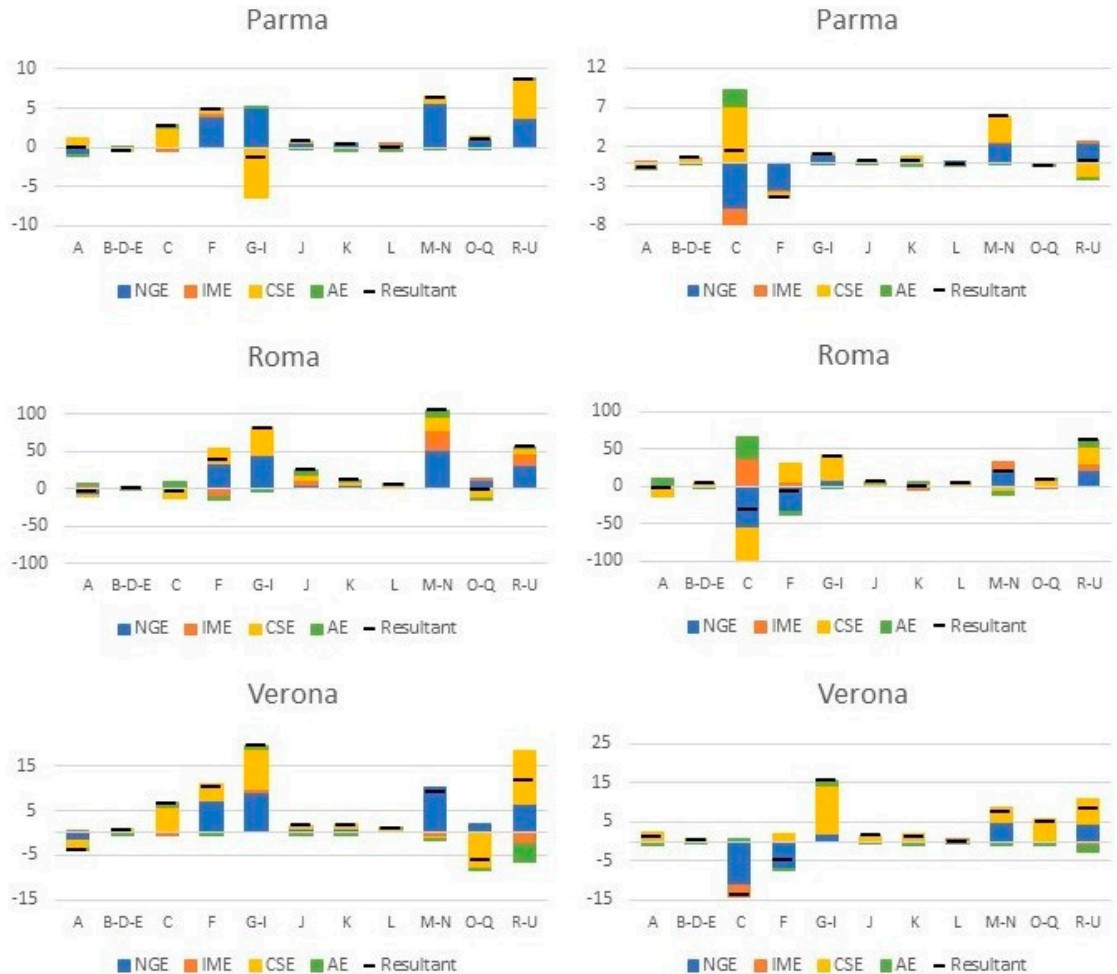

**Figure 6.** Shift-share effects of the sectoral employment of the strong resilient Italian metro-regions: Bologna, Firenze, Genova, Milano, Padova, Parma, Roma, and Verona. Periods: 2000–2008 (**left**), 2008–2016 (**right**). The *x*-axis represents the codes of the economic sectors, as specified in Appendix A.

The sectors of Agriculture (a), Non-Manufacturing Industry (b-d-e), Information and Communication (j); Finance and Insurance (k), and Real Estate (l) show a trend that is not specific to this group but is similar in all the Italian metro-regions, both resilient and vulnerable. In these sectors, the crisis had the result of "freezing" the previous occupational dynamic (if any and of any sign), determining in the 2008–2016 period limited effects, mainly of the type competitive (CSE) and allocative (AE), which have an opposite sign and compensate each other. If we exclude the increases in ICT employment in Milano (+11,800) and Roma (+6100), the largest increase was in Bologna of around +4300 employees in the same sector. In the remaining sectors, it is difficult to find a typical behaviour. The Manufacturing sector (c) has a widely negative national effect (NGE) that adds up with different combinations of other negative effects: in the case of Roma, a significant contribution comes from the competitive effect (CSE); in the case of Firenze, from the combination of the allocative (AE) and the competitive effect (CSE); in the case of Verona, Padova, Parma, Bologna, and Genova from the industrial mix effect (IME). In all these cases the final result is the same: a loss in Manufacturing employment, which ranges from −56,700 employees in Milano to −7800 in Genova. Finally, the sector of Cultural and other Services (r–u), often driven by the competitive effect (CSE), plays a vital role in the final employment rate, in cases of both positive and negative signs.

### 3.3. Weak Resilient Italian Metro-Regions

In the graph of Figure 5, the group of the weak resilient regions corresponds to the light blue sector. This group includes the Metropolitan cities of Venezia and Torino and the northern, medium-sized metro-region of Bergamo. As to the occupational capacity, the information represented by the spheres shows it is positive in Bergamo (+19.9%), quite positive in Venezia (+8.2%), highly negative in Torino (−42.3%). This is due to the different occupational dynamic these metro-regions had when the crisis started.

Figure 7 depicts the post-crisis dynamics, showing common effects mainly in the following sectors:

- Construction (f): in this sector the crisis had the effect of turning the previous positive values of the national, industrial and competitive effects into negative values, of almost the same intensity, determining a loss of employment, not compensated by the fact that the regional labour market could perform better than the national one. In the case of Torino and Bergamo, in particular, this advantage with respect to the nation (i.e., a positive value of the CSE) appears only after 2008. In the case of Venezia, it is already present in the pre-crisis period.
- Technical and Scientific Services (m–n): in this sector, the crisis had determined a resize of employment, imputable above all to the national and competitive effects, that did not, however, change the positive sign of the occupational dynamics. In all the three regions, the period 2008–2016 ends with an increase in employees that amounts to +18,800 in the case of Torino, +3300 in Bergamo, and +4400 in Venezia.

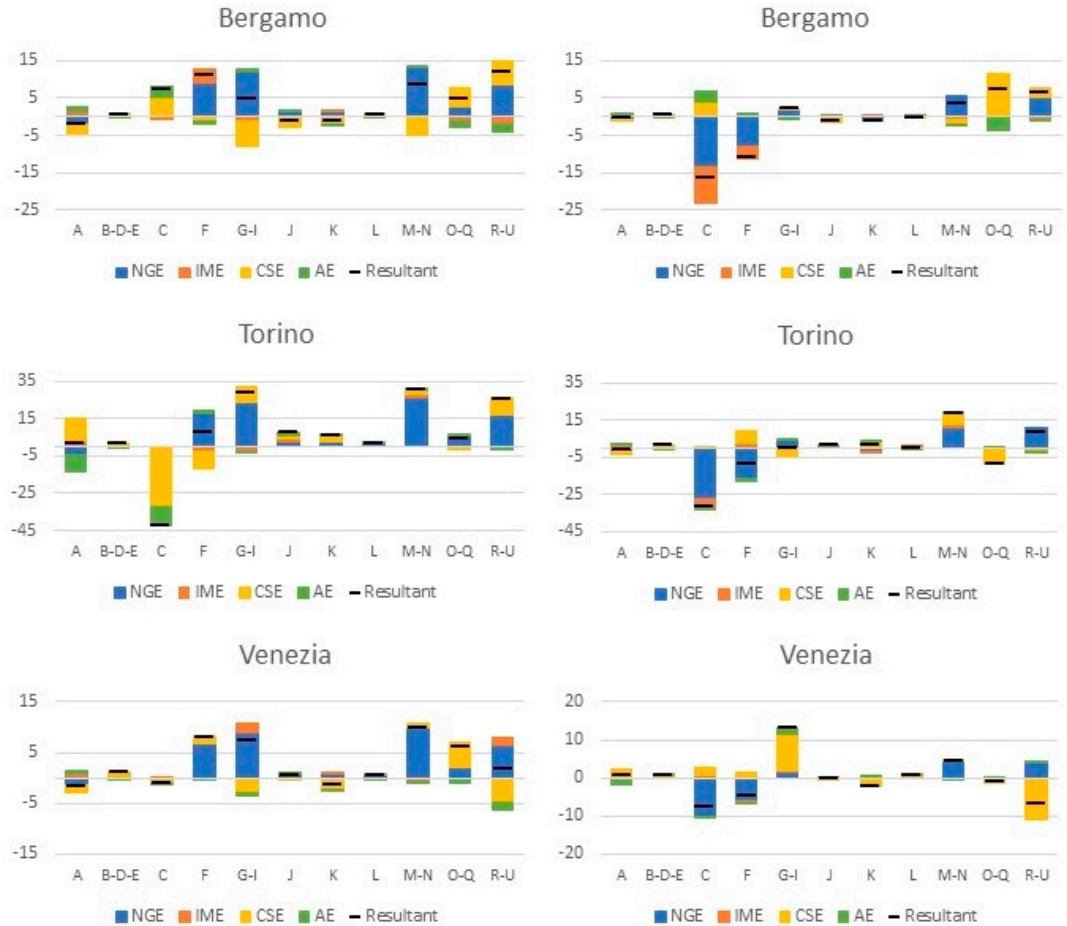

**Figure 7.** Shift-share effects of the sectoral employment of the weak resilient Italian metro-regions: Bergamo, Torino and Venezia. Periods: 2000–2008 (**left**), 2008–2016 (**right**). The *x*-axis represents the codes of the economic sectors, as specified in Appendix A.

In the Retail and Logistic sector (g–i), the situation of the three regions varies greatly both in the final balance of the pre-crisis and post-crisis periods, and in the consistency of the different effects. In Venezia, the fading away of the positive contribution given by the national effect is compensated by a strong increase in the competitive effect, which raises the employment in the sector of around 13,200 units. In Bergamo, this balance amounts to +2100, while in Torino it reaches the negative value of −500. In addition, in the sector of the Public Administration services (o–q) and the sector of the Cultural and other Services (r–u), a common trend does not seem to be identifiable. While in the remaining sectors, we notice the same "levelling" dynamic observed for the strong resilient metro-regions, that sets to zero the variation of employment in the 2008–2016 period. The most extensive variation for all these sectors is the one of Venezia: −2100 employees in Finance and Insurance (k), pulled down by a negative competitive effect (CSE).

### 3.4. Not Resilient, Reactive Italian Metro-Regions

This group, depicted in the purple sector of Figure 5, includes just two metro-regions: Prato in the centre of Italy, and Bari in the South. With such a limited sample, it follows that the identification of a common trend has to be taken with caution. As to the occupational capacity, these two regions are both characterized by a very negative situation (quantified, respectively, in −70.1% and −115.7%), which emerges as structural, as a negative occupational balance was already present in the years before 2008.

These regions emerge from the rest of the sample for the presence of an ambiguous condition of vulnerability. This condition is characterised by a low occupational capacity (testified by the size and negative value of the spheres in the graph of Figure 5) and an occupational performance always weaker than the national one, but with a certain reaction to the crisis.

Again, in this group, the sectors of Agriculture (a), Non-Manufacturing Industry (b-d-e), Information and Communication (j); Finance and Insurance (k), and Real Estate (l) tend to give a residual contribution to employment (see Figure 8), while in all the remaining sectors, it is not possible to distinguish a common dynamic. We interpret this result as the clue (still to be tested) that, differently from resilience, the reactiveness relies above all on the specific conditions that characterise the region before and after the crisis.

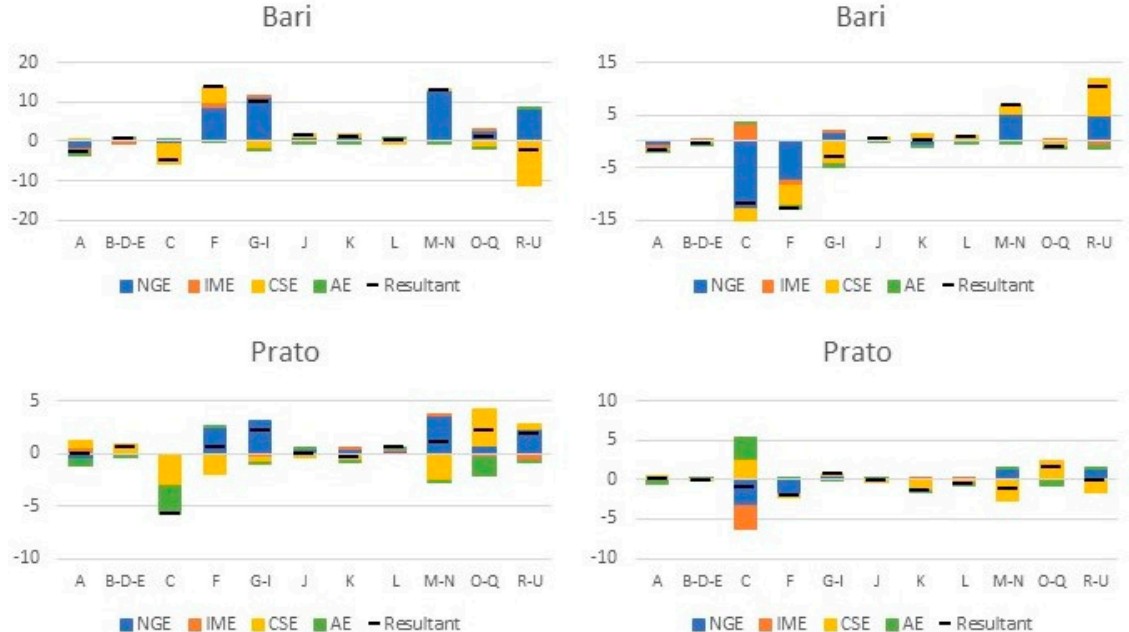

**Figure 8.** Shift-share effects of the sectoral employment of the *not resilient*, *reactive* Italian metro-regions: Bari and Prato. Periods: 2000–2008 (**left**), 2008–2016 (**right**). The *x*-axis represents the codes of the economic sectors, as specified in Appendix A.

### 3.5. Not Resilient, Vulnerable Italian Metro-Regions

In Italy, the group of the vulnerable metro-regions, represented by the grey sector of Figure 5, is the most crowded of all but paradoxically a bit less heterogeneous. From a geographical point of view, with the only exception of the metro-regions of Brescia and Reggio Emilia, all the regions falling in this group are metro-regions of the south of Italy: Cagliari, Catania, Messina, Napoli, Palermo and Taranto (see Section 4). Secondly, what distinguishes mostly these regions from the resilient and reactive ones is an occupational dynamic that is weaker than the nation in all the considered variables with the only exception of the occupational capacity of Palermo, Cagliari, and Reggio Emilia, which is positive and higher than the average. Conversely, Napoli and Messina are the most vulnerable of the group. While in the remaining cases, the loss of employees is mainly limited, also compared to the national average. As to the variable of the occupational capacity, situations are highly diversified, with values that range from the highly positive value of Palermo (+91.8%) to the highly negative one of Napoli (−100.1%).

Some further considerations on the sectoral response to the crisis derive from the graphs shown in Figure 9. As noted for other types of resilience, also the regions of this group are characterised by residual post-crisis effects in Agriculture (a), Non-Manufacturing Industry (b-d-e), Information and Communication (j), Finance and Insurance (k), and Real Estate (l). The Agriculture sector in Napoli is the only exception, registering a final balance of employees of around −8.200. In the other sectors, the crisis resulted in a "turmoil effect", producing a generalised amplification of the magnitude of all the effects. The sectors Manufacture (c), Construction (f) and Retail and Logistic (g–i) usually end the 2008–2016 period without a loss of employment or, at their best, with a residual increase. Only Taranto registers an increase in a certain relevance, that amounts at around +3800 employees in the sector of Retail and Logistic.

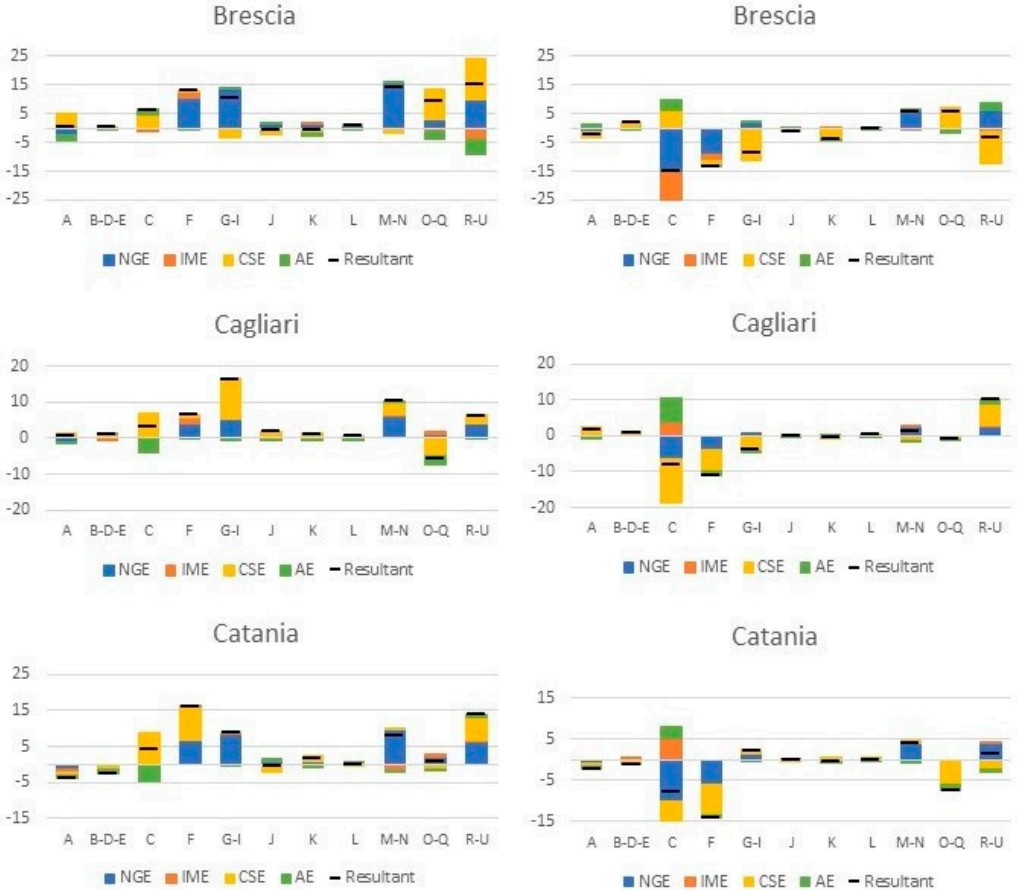

**Figure 9.** *Cont.*

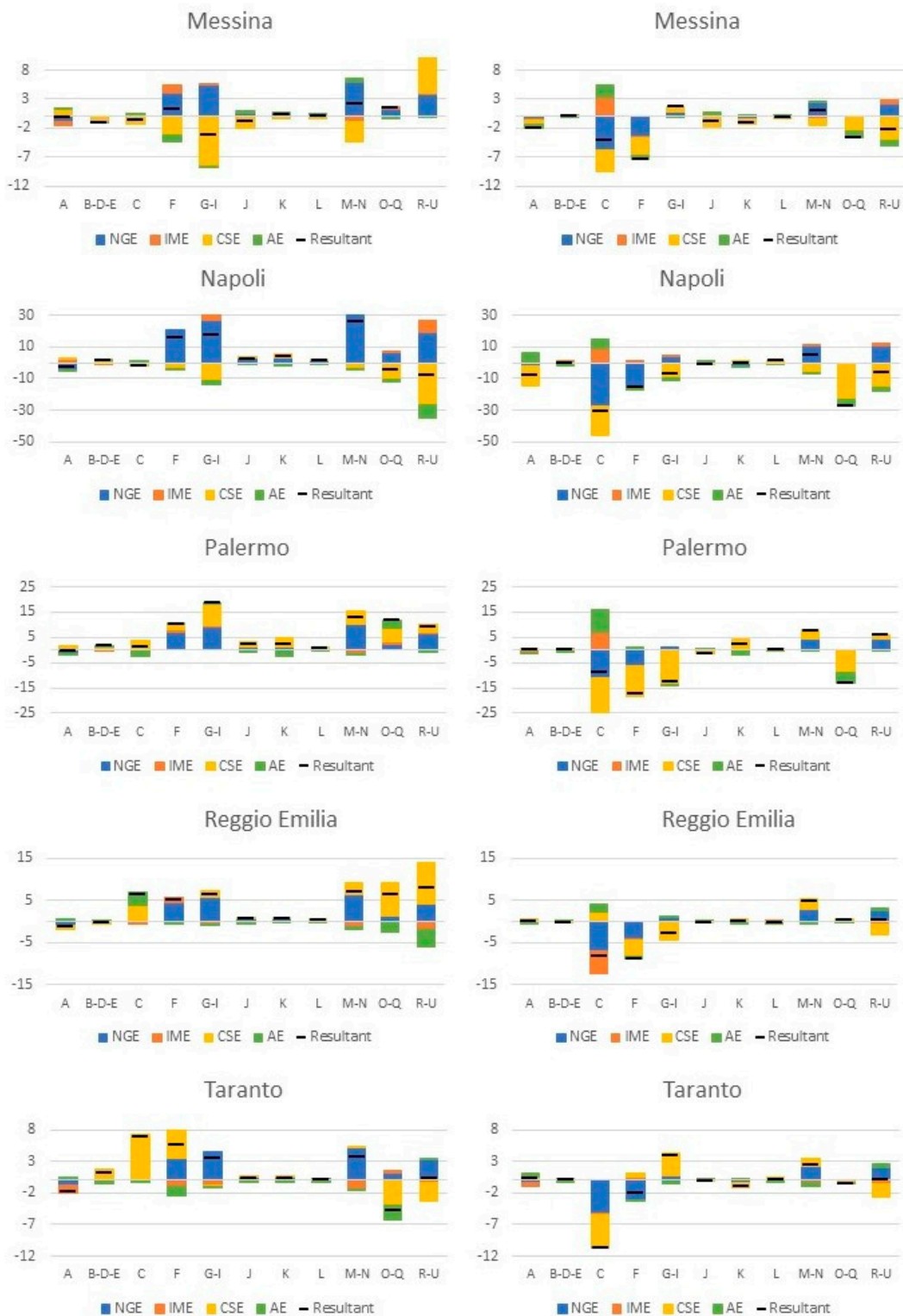

**Figure 9.** Shift-share effects of the sectoral employment of the *not resilient*, *vulnerable* Italian metro-regions: Palermo, Cagliari, Reggio-Emilia, Brescia, Catania, Taranto, Napoli, Messina. Periods: 2000–2008 (**left**), 2008–2016 (**right**). The *x*-axis represents the codes of the economic sectors, as specified in Appendix A.

## 4. Discussion

The paper proposes an analysis exploring a different way to construct regional taxonomies of resilience. In doing that, it refers to the consistent bulk of literature that deals with economic resilience,

and assumes the employment variable, as it provides useful hints from both an interpretative and methodological perspective.

Indisputably, employment is not the only variable representative of regional resilience. We are aware it is just a proxy of a specific type of resilience, which is economic resilience. Other types of resilience, also inclusive of the social, institutional and environmental dimensions of growth, the influence of territorial spillovers [32] and the occurrence of international effects [57], would have requested not just a variable but a multidimensional set of variables [34,36,40]. Resting on the economic literature, we are also aware that some more variables could have been explored, such as the value-added or the gross domestic product [38,58]. However, we intended this paper as a preliminary study, open to be tested with other data and other types of resilience. In such a perspective, the employment variable has the advantage of representing both economic and social issues. It is, in fact, a measure of the human capital of a region and its level of specialisation/diversification. Furthermore, it allows for a regional and sectoral disentanglement of a dynamic of growth, which is distinctive of the shift-share methodology and allows, in a non-deterministic way, one to describe how the different sectors contribute to distinguishing the regional path from the national one.

The analytical instrumentation adopted in this study allows one to highlight the economic resilience of the Italian metro-regions to the 2008 crisis. Moreover, it allows one to formulate some hypotheses on the influence of the economic sectors on the reactiveness of metro-regions. In the geographical and regional literature, this is a critical issue. Economic specialisation is considered a crucial factor in the explanation of economic growth and innovation. Employment in high-tech industries and services goes on being considered a vital factor for the regional economy as businesses in these sectors keep on playing a particularly important role in pushing economic attractiveness and introducing new products and services that impact the entire economy.

From this point of view, however, the results of the shift-share we developed are not conclusive. The combination in time and space of the shift-share effects shows that the main common emergent trait of the resilient regions is the persistence, before and after the crisis, of a positive competitive capacity (CSE). At the level of the Italian economy, the negative effects of the crisis are evident above all in the national effects (NGE) of the Manufacturing and Construction sectors. The Construction sector is also the branch of the economy that contributes most—together with the sector of Technical and Scientific Services—to differentiate the resilient regions, both strong and weak, from the not resilient ones.

The second point of discussion with implications on the geographical and regional debate regards the representation of the economic performances of the Italian territories. The taxonomic analysis of regional resilience we developed with reference of Italian metro-regions, in fact, draws a geographic representation of the leading economies at the national scale (represented in the map of Figure 10) that is only partially consistent with the most popular representations of the economic growth imbalances within the country.

As reported in Table 2, if we consider the most famous "visions" of the national economy produced since the 60 s (for a review, see the interesting work of Bartolini [59]), we find that only the North–South divide shows a good correspondence with the results of our analysis. Indeed, with the only exception of Brescia and Reggio Emilia, all the vulnerable metro-regions are southern regions, and all the resilient ones are northern or central regions. Roma is the most southern region in the group of the strong resilient.

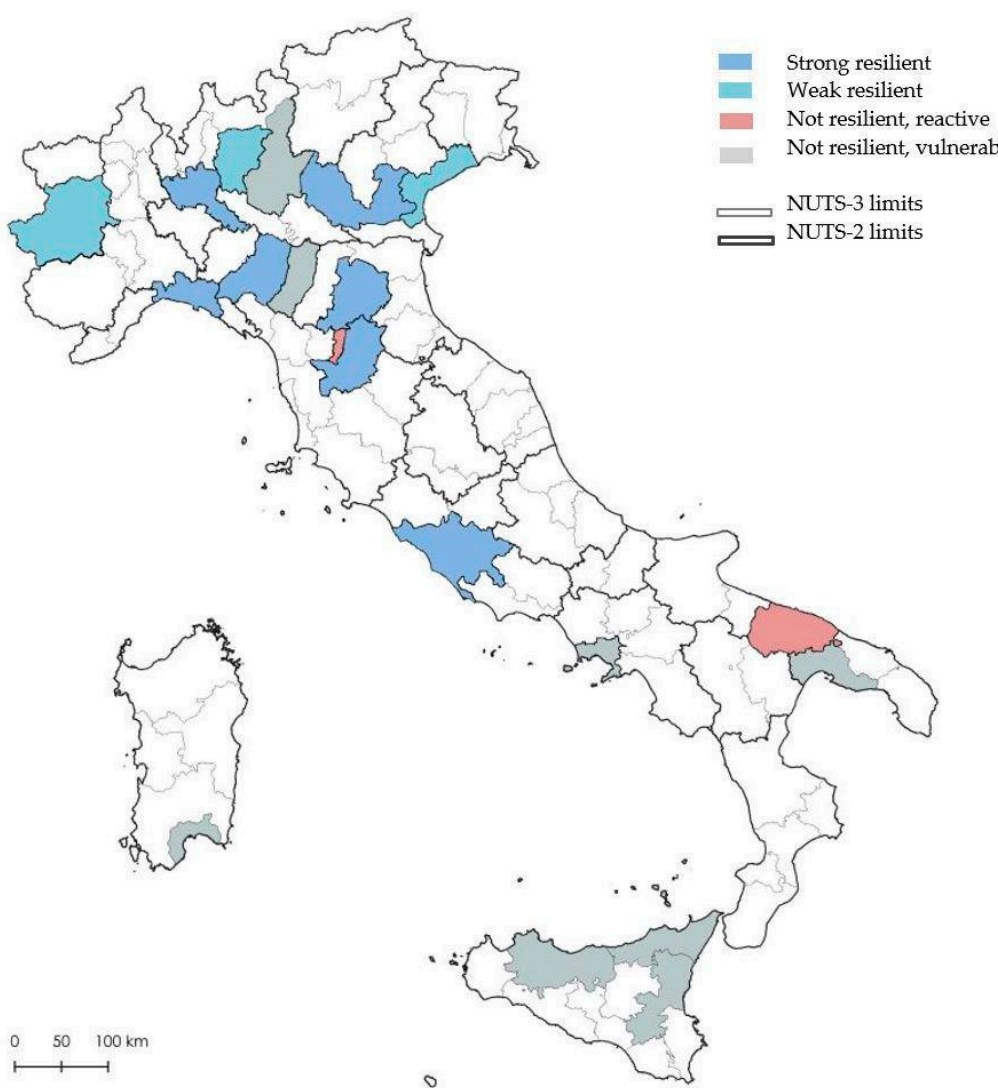

**Figure 10.** The economic resilience to the 2008 crisis of Italian metro-regions.

**Table 2.** The consistency of the proposed taxonomy of resilience with some main representations of the Italian economic divides.

| Representations | Consistency | Analogies | Divergences |
| --- | --- | --- | --- |
| North–South divide [60] | High | Milano, Genova, Verona, Padova Parma, Bologna, Firenze, Roma, Venezia, Torino, Bergamo are core/resilient. | Brescia and Reggio Emilia are vulnerable. Bari is reactive, although not resilient. |
| Third Italy/North–East–Centre Marshall economies [61,62] | Low | Verona, Padova, Bergamo are core/resilient. | Brescia and Reggio Emilia are vulnerable. Prato is reactive, although not resilient. |
| Metropolitan cities as national champions [63] | Low | Milano, Torino, Genova, Bologna, Firenze, Roma, Venezia, Torino are core/resilient. | Napoli, Messina, Catania, Cagliari, Palermo are vulnerable. Bari is reactive, although not resilient. |
| Large urban economies (core cities >500,000 inhabitants) [64] | Moderate | Roma, Milano, Torino, Genova are core/resilient. | Napoli and Palermo are vulnerable. |

Source: authors' elaboration.

The correspondence is very poor, instead, concerning both the "rhetoric" of the national champions assumed by the Italian reform of the local entities (Delrio Reform) and the conceptualisation of regional growth as a function of the urban rank/competitiveness. Eight of fifteen Italian Metropolitan cities—which the Delrio Reform entrusted to push the overall national economy forward [63]—present features of resilience. The residual ones except for Reggio Calabria, not included in the analysis as not inserted in the list of the metro-regions by Eurostat) are vulnerable. Even less are the metro-regions that are consistent with the so-called "Third Italy" or "NEC" (North–East–Centre) territories, characterised by a prolonged presence of Marshallian economies.

Finally, the Gramscian vision of the role of large urban economies results partly coherent with our results. If we consider the population of the core cities of each metro region, we see that four of the six largest cities in Italy (>500 thousand inhabitants) are part of a resilient region. However, the evaluation of this last representation is biased by the criteria used to select the cases analysed, as population size is one of the criteria used by Eurostat to identify the European metro-regions.

Table 2 proposes a synthetic, not-exhaustive description of the consistency of the classification of the Italian metro-regions with some well-established representations of the economic divides within the country. As the table shows, there are divergences concerning all the representations. In addition, among the cases listed in the column of the analogies, slight differences derive from the fact that we distinguished the strong resilient from the weak resilient ones. It is the case, for instance, of Torino and Venezia, which emerge with a feeble resilience compared to other metro-regions similar for population or economic size (Milano and Bologna for Torino, Padova for Venezia; see Table A2 in Appendix C). How to explain these discrepancies and differences? A first option is that the discrepancies derive from the methodology adopted and the choice to measure resilience only via the employment growth rate and the 11-sector employee distribution. However, our results are also consistent with the evidence resulting from place-specific studies and analysis.

Going back to Torino and Venezia, in the first case, a relevant number of studies have identified a process of golden decline determined by the post-Fordist restructuration of the local and the global economy [65,66]. In the case of Venice, the transition towards post-industrial forms of economic development has been poorly managed by the local and national politics [67] and a lack of a comprehensive strategic vision has been produced, namely, regarding the old industrial areas still present in the metro-region [68]. Brescia and Reggio Emilia suffered both from the global economic crisis and other unfavourable events (such as the 2012 earthquake in Reggio Emilia) in a way that mined the competitiveness of the local labour market. A relevant bulk of literature [30,63,65,66] also testifies the failure of the majority of the Metropolitan cities created by the Delrio Law in being levers of economic growth (including those in row 3 of Table 2). In such a context, Bari (quoted in rows 1 and 3 in Table 2 as reactive), however, emerges as a large Metropolitan city of the south of the country that had the benefit in recent years of two important pushing conditions: the availability of public funding from the EU structural funds and some important national programs (such as the Agenda Digitale and Smart Cities programs) and the presence of effective governance processes [69].

A second option is thus that some of the most renowned geographies of the Italian divides—often elaborated before the crisis started—have at least partly changed.

Besides confirming the importance of an evolutionary, path-dependent approach to the analysis of economic resilience, this result is quite intriguing and paves the way for further research. In the reflection on economic development, there are at least two other important representations that it may be worth testing with an analysis of the type we proposed: the small places' dynamism and the revenge of the "forgotten" places. Both these representations recognise economic centrality to the marginal, the rural and the inner areas, recognising them trajectories of growth alternative with respect to a neoliberal technology-led development model. The former, particularly, emphasises the capacity of small villages to allow healthier lifestyles and exploit the opportunities of economic growth of an ageing (silver) society [58,70]. The latter supports the idea of an uneven centrality of marginal cities and regions, often ignored by the national policy as "places that don't matter" [71].

The focus on metro-regions does not allow for the consideration of small- and medium-sized economic systems. The smallest metro-region in Italy corresponds to the province of Prato, which hosts 255 thousand inhabitants, while the national average is 550 thousand inhabitants per province. However, it might be highly instrumental in testing the resilience of all the subnational regions forming the national economy using, for instance, the research unit of the local labour systems. In such a perspective, the National Statistical Agency of Italy (Istat) has recently released a report entitled "*Rapporto sul Territorio*" that provides some interesting clues to read the changing national economy.

**Author Contributions:** Conceptualization, F.S.R., M.B. and P.F.; Methodology, F.S.R., M.B. and P.F.; Validation, F.S.R., M.B. and P.F.; Formal Analysis, M.B. and P.F.; Investigation, F.S.R., M.B. and P.F.; Data Curation, F.S.R. and P.F.; Writing-Original Draft Preparation, F.S.R. (Section 1 "Introduction", Section 3 "Results" and Appendix A), M.B. (Section 2 "Materials and Methods" and Appendix B) and P.F. (Section 4 "Discussion" and Appendix C); Writing-Review & Editing, F.S.R. (Section 1 "Introduction", Section 3 "Results" and Appendix A), M.B. (Section 2 "Materials and Methods" and Appendix B) and P.F. (Section 4 "Discussion" and Appendix C); Visualization, F.S.R. and P.F.; Supervision, M.B. All authors have read and agreed to the published version of the manuscript.

**Funding:** This research received no external funding.

**Conflicts of Interest:** The authors declare no conflict of interest.

## Appendix A

**Table A1.** Correspondence between the economic sectors used in the shift-share analysis and the Nomenclature of Economic Activities NACE rev.2 codes.

| Sectors | NACE Rev. 2 |
| --- | --- |
| Agriculture | A: Agriculture, forestry and fishing |
| Manufacturing | C: Manufacturing |
| Non-Manufacturing Industry | B–D–E: Mining and quarrying; Electricity, gas, steam and air conditioning supply; Water supply; sewerage, waste management and remediation activities |
| Construction | F: Construction |
| Retail and Logistic | G–I: Wholesale and retail trade; repair of motor vehicles and motorcycles; Transporting and storage; Accommodation and food service activities |
| Information and Communication | J: Information and communication activities |
| Finance and Insurance | K: Financial and insurance activities |
| Real Estate | L: Real estate activities |
| Technical and Scientific Services | M–N: Professional, scientific and technical activities; Administrative and support service activities |
| Public Administration Services | O–Q: Public administration and defence, compulsory social security; Education; Human health and social work activities |
| Cultural and other Services | R–S–T–U: Arts, entertainment and recreation; Other services activities; Activities of households as employers; Activities of extraterritorial organisations and bodies |

## Appendix B

The idea of the occupational capacity originates from the cumulative formulation of the shift-share adopted in this study. Using the conceptual framework of Equation (5), we propose the introduction of the absolute occupational capacity of region $r$ and sector $i$, $AOC_{ir}{}^{t_0-t_{0+h}}$, defined over the interval between year $t_0$ and year $t_{0+h}$ and calculated as the summation of all the cumulative sum introduced in Equation (5) for every year of the time period, as expressed by:

$$AOC_{ir}{}^{t_0-t_{0+h}} = \Sigma_n CS_{ir}{}^{t_0-t_{0+n}} = CS_{ir}{}^{t_0-t_{0+1}} + CS_{ir}{}^{t_0-t_{0+2}} + \ldots + CS_{ir}{}^{t_0-t_{0+h}} \tag{A1}$$

where *n* varies between 1 and *h*.

To better clarify the metrics introduced in this paper, we consider the exemplificative case study of the metro-region of Roma. Figure A1 (left) shows, for every year of the interval 2000–2016, the histograms representing the four shift-share effects calculated with expressions (3) and (4). These equations contain information on the employment variation of each single year with respect to the previous one, but do not say anything about the global final results. The cumulative sum, defined in Equation (5) and reported in Figure A1 (right), provides the desired information, because it calculates, for each year, the result of the partial summation.

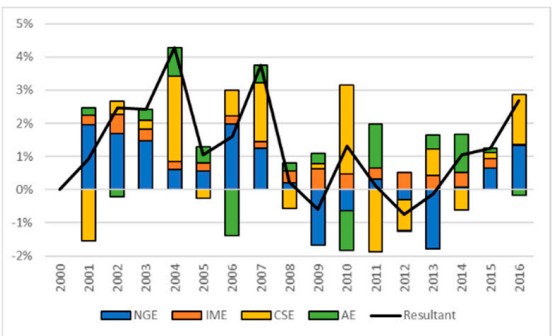 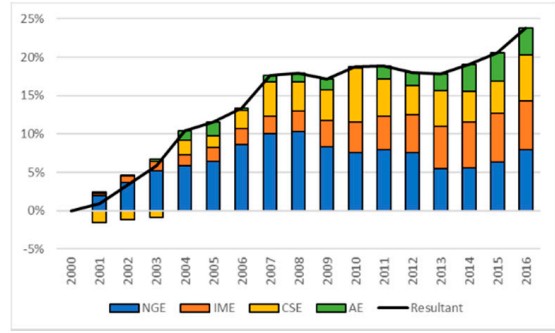

**Figure A1.** The exemplificative case study of the metro-region of Roma. Shift-share effects calculated with expressions (3) and (4) (**left**), and their cumulative sum, defined in Equation (5) (**right**). Black lines indicate the resultants.

The absolute occupational capacity, defined by the summation of Equation (A1), represents, from a mathematical point of view, the discrete version of the integral of the cumulative sum (the curve reported in Figure A1 right).

To focus on the differences between regions and the nation, we introduce the definition of the *relative occupational capacity*, $ROC_{ir}^{t_0-t_{0+h}}$, calculated as the difference between the absolute occupational capacity of the region and the absolute occupational capacity of the nation:

$$ROC_{ir}^{t_0-t_{0+h}} = AOC_{ir}^{t_0-t_{0+h}} - AOC_{iN}^{t_0-t_{0+h}} \tag{A2}$$

where *N* indicates the national value.

**Appendix C**

**Table A2.** Population and employment of Italian metro-regions in 2016.

| Metro-Region [1] | Population | Employment |
|---|---|---|
| Bari | 1,263,820 | 471,100 |
| Bergamo | 1,108,298 | 483,400 |
| Bologna | 1,005,831 | 518,100 |
| Brescia | 1,264,105 | 555,900 |
| Cagliari | 561,289 | 230,300 |
| Catania | 1,115,535 | 352,300 |
| Firenze | 1,013,348 | 508,600 |
| Genova | 854,099 | 393,200 |
| Messina | 640,675 | 200,400 |
| Milano (Provinces of Milano, Lodi, Monza Brianza) | 4,303,998 | 2,347,600 |
| Napoli | 3,113,898 | 981,400 |
| Padova | 936,887 | 445,000 |
| Palermo | 1,271,406 | 375,500 |
| Parma | 447,779 | 223,200 |

**Table A2.** *Cont.*

| Metro-Region [1] | Population | Employment |
|---|---|---|
| Prato | 253,123 | 121,800 |
| Reggio nell'Emilia | 532,872 | 242,400 |
| Roma | 4,340,474 | 2,127,300 |
| Taranto | 586,061 | 188,700 |
| Torino | 2,282,197 | 988,200 |
| Venezia | 855,696 | 371,600 |
| Verona | 922,383 | 426,800 |

[1] All the Italian metro-regions, except Milano, correspond to the NUTS II level and the Province/Metropolitan city level. In the case of Milano, the metro-region results from the sum of three NUTS II: Milano (which is also a Metropolitan city), Lodi and Monza Brianza (Provinces).

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
