# Peer review of "Breaking the Black-Box of Regional Resilience: A Taxonomy Using a Dynamic Cumulative Shift-Share Occupational Approach"

_sustainability, doi:10.3390/su12219070_

Round 1

Reviewer 1 Report

Very interesting and important topic and creative taxonomy proposition. It is a novel contribution to the resilience literature and as such, it is very remarkable. 

However, the study needs to be improved by rewording it more accurately at several points:

Comments:

Line 54: What do these numbers in brackets ... (1, 2, 5b, and 6) mean?

Line 61:  Wignell, Martin and Eggins [ needs to be erased

Lines 62-68: please revise, same thought appears twice.

Lines 85-87: Please consider revising the sentence is too complicated, I don't understand

Lines 92-122: This part isn't clear enough for me. Some 'taxonomies' are here types of the concept resilience, some are typologies of regions. In my opinion these are concepts related to resilience, but not taxonomies.

Lines 154-156: Needs a verb

Lines 191-192: Sentence is not clear

It would be useful to list the three steps of the taxonomy generating process in advance, before line 200.

 Lines 205- 206: The formula and description is inaccurate. Martin describes the calculations much more detailed, which I miss here. (Employment change is indicated here instead of percentage change in employee number)

Line 217: 'the total amount of workplaces generated by the region' this part of the sentence suggests that it is about the change in number of workplaces, although description of calculation says that it is the amount and not the change. Maybe a formula could clarify it.

Lines 254-275: Please consider revising in terms of definitions being independent from each other (do not refer to definitions of other types)

Lines 323-325: Consider revising, false resilient, persistent regions - based on the figure - being in upper-right sector means only relatively positive performance, not  'positive increases of the employment' in both periods. And also, we don't know where the horizontal dashed line is.

Line 330: Left to the diagonal: it should be written right

Line 362: In the formula E*irt0 there is a typo: E*irt0=(Eirt0)(... i should not be there.

Lines 370-371: Please consider leave this out: measures the degree of similarity of the region to the national reality,

In the results part,

  • Question is: Applies the national average to the whole country or only to average of metro regions?
  • It is not clear, what occupational performance means because of lack of formulas in the methods part.
  • please indicate industries with capital letters
  • give higher quality figures.

Discussion summarizes the findings very well, although it should give us more feedback about the taxonomy proposition itself.

Reviewer 2 Report

The paper is well structured and relevant for the topic and the proposed approach.

Concerning the applied methodology, some general remarks:

  • "The first step: using the sensitivity index to propose a new taxonomy": i really don't think that the regional trends in employment could be the only indicators representative of the whole regional resilience bahaviour. So it is very import to document why you chose this way simplifying may be to much the very complex feature we call "resilience"
  • "The quantification of the sectoral influence on resilience", in this stage from the total employment you move toward a sectorial data combination of economic sector contribution, so you affirm more strongly the relation between employment and resilience. This should be argued in a critical way, not only proposed as a solution for the proposed classification method. 

The proposed taxonomy represents a meaningful classification (fig. 4) of regional classes compared with resilient attribute.

Discussion of results: the classification of the Metropolitan Cities of Venezia and Torino as "Weak resilient Italian metroregions" is not really comfortable and probably reveal the weaknesses of the adopted approach. The question is: how do you test the results? can you recognize economic resilience evidences in your classification according with 

Please improve the quality of FIG 6 - it is not fully readable.

I recommend to be more clear in the conclusions: "The analytical instrumentation adopted in this study allows highlighting the economic resilience (or vulnerability) of the Italian metroregions to the 2008 crisis." 

According also to your point of view resilience is not vulnerability. This association keeps the reader in confusion. The same at line 562.

Author Response

Kind regards

Reviewer 3 Report

Out of format citation:

61  exercise with the “experiential world” [Wignell, Martin and Eggins [1].

  1. Martin et al.

  2. 385  2016, for instance, run a fine-tuned 25-sector disaggregation. Artige and van Neuss 2014

In most cases?

  1. 62  In most of the cases,

  2. 65  In most of the cases,

Repeated paragraph:

  1. 62  In most of the cases, however, taxonomic practices are either too sophisticated (for instance

  2. 63  relying on elaborate forms of cluster analysis) or endowed with excessive selectivity related to the

  3. 64  context of the analysis and the availability of data [4].

  4. 65  In most of the cases, however, taxonomic practices are either too sophisticated - utilising, for

  5. 66  instance, complex procedures of cluster analysis - or characterised by excessive selectivity related to

  6. 67  the context of the analysis and the availability of data [4]. It happens in the study of economic

  7. 68  resilience, too, with some peculiarities.

Reference?

  1. As Nijkamp, Zwetsloot, and van der

  2. 71  Wal have observed, in the XXth century regional and urban discourse,

Double not?

95

References

Typologies

Resistant, Recovered, Re-orientated, Renewed

Resistant, Recovered, Not recovered in upturn, Not recovered not in upturn

The?

145  so complex and the diverse

Aim? The aim?

  1. 165  Aim of the paper

The absolute? The relative?

  1. Because we do not consider the

  2. 211  absolute but the relative economic performances,

Shouldn't Figure 5 be Figure 2 (or split in two)?

Diriment?

  1. 609  From this point of view, however, the results of the shift-share we developed are not diriment.

Author Response

Kind regards

Round 2

Reviewer 2 Report

The paper improved according to requested clarification.

The general structure is acceptable for publication.

Minor english check has to be done